# FINDING SPARSE AUTOENCODER REPRESENTATIONS OF ERRORS IN CoT PROMPTING

**Justin Theodorus, V Swaytha, Shivani Gautam, Adam Ward, Mahir Shah, Cole Blondin, Kevin Zhu**

Algoverse AI Research

## ABSTRACT

Current large language models often suffer from subtle, hard-to-detect reasoning errors in their intermediate chain-of-thought (CoT) steps. These errors include logical inconsistencies, factual hallucinations, and arithmetic mistakes, which compromise trust and reliability. While previous research focuses on mechanistic interpretability for best output, understanding and categorizing internal reasoning errors remains challenging. The complexity and non-linear nature of these CoT sequences call for methods to uncover structured patterns hidden within them. As an initial step, we evaluate Sparse Autoencoder (SAE) activations within neural networks to investigate how specific neurons contribute to different types of errors.

## 1 INTRODUCTION

Large language models (LLMs) have demonstrated remarkable capabilities when generating step-by-step reasoning or chain-of-thought (CoT) responses (Wei et al., 2022). However, these intermediate reasoning traces often contain subtle, hard-to-detect errors ranging from logical inconsistencies to factual hallucinations, arithmetic slips, and more (Tyen et al., 2023). Such polysemanticity—where a neuron activates in multiple semantically unrelated contexts—complicates attempts to interpret or debug these models' internal states (Haider et al., 2025).

While prior work on mechanistic interpretability has produced insights into how LLMs represent features internally (Molnar, 2024), a structured approach to detecting and categorizing reasoning errors remains elusive (Yeo et al., 2023). In particular, existing interpretability methods often focus on uncovering how to improve final model output, rather than on systematically investigating errors within CoT steps. This gap motivates us to leverage Sparse Autoencoders (SAEs), which can learn individual "directions" or features in a model's activation space and have demonstrated success in reducing polysemanticity (Olsson et al., 2023).

In this paper, we present the following preliminary approach:

1. **Labeling CoT Errors:** Constructing a labeled dataset of 1,000 chain-of-thought (CoT) answers from the GMS8K dataset, categorizing them into nine distinct error types.

2. **SAE Activation Analysis:** Extracting sparse autoencoder (SAE) activations from a 2B-parameter model (Gemma 2B) to determine which neurons or feature directions align with specific error patterns.

3. **Correlation Exploration:** Investigating correlations between sparse features and error types, uncovering potential relationships between model activations and reasoning failures.

By identifying which sparse features co-activate with particular error categories, we aim to provide insights into how reasoning failures manifest in activation space, contributing to more interpretable chain-of-thought debugging.

Given the novelty of applying Sparse Autoencoders in the field of interpretability research, our primary goal of this paper is to introduce our methodology and highlight key research directions. Ultimately, we aim to provide a stepping stone for more interpretable chain-of-thought debugging, helping both researchers and practitioners build more trustworthy LLMs.

## 2 BACKGROUND AND RELATED WORK

### 2.1 SPARSE AUTOENCODERS FOR INTERPRETABILITY

Recent work demonstrated that polysemantic neurons in LLMs can be decomposed into more monosemantic directions via sparse reconstruction. Their method shows that superposition can be alleviated, enabling clearer interpretability of internal network states (Cunningham et al., 2023), providing a foundation for transparent language models.

### 2.2 AUTOMATIC EVALUATION OF REASONING STEPS

"ROSCOE: A Suite of Metrics for Scoring Step-by-Step Reasoning" introduces unsupervised scores to evaluate the correctness of rationales independently from the final answer. These unsupervised scores aim to measure semantic consistency and factuality of reasoning steps. However, ROSCOE primarily focuses on end-to-end correctness rather than in-depth interpretability or correlation with internal neuron-level features (Golovneva et al., 2023).

### 2.3 ENHANCING SPARSE AUTOENCODERS WITH FEATURE ALIGNMENT

Recent work has explored regularization techniques such as Meaningful Feature Representation (MFR) to improve feature learning in SAEs, making them more likely to uncover meaningful input features (Marks et al., 2024). Their work further validates the potential of SAEs for uncovering interpretable patterns in network activations across domains, from EEG signals to GPT-2 text models (Marks et al., 2024). This further validates the potential of SAEs for uncovering interpretable patterns in network activations.

## 3 METHODOLOGY

### 3.1 GSM8K DATASET AND ERROR TAXONOMY

The error taxonomy we use in our study is derived from the ROSCOE set of metrics (Golovneva et al., 2023), which provides a structured framework for categorizing reasoning errors in chain-of-thought (CoT) prompting (See Table 1 for definitions of error types). The ROSCOE taxonomy was developed through an extensive analysis of reasoning failures in language models, allowing us to systematically classify errors in CoT answers.

In our study, we select 1,000 CoT answers from the GSM8K dataset, generated by the Gemma-2b language model. Each answer is manually annotated into one or more of the error categories defined in ROSCOE.

Table 1: Error Type Definitions, (Golovneva et al., 2023)

| Error Type | Definition |
| --- | --- |
| **Grammar** | Use of incorrect, unconventional, or controversial grammatical structures. |
| **Factuality** | Details about objects are inconsistent with the provided context. |
| **Hallucination** | Information that is not present in the input and is either incorrect or irrelevant. |
| **Redundancy** | Includes redundant details that, while factually accurate, do not contribute to answering the question. |
| **Repetition** | Restating previously mentioned content in different wording within the reasoning steps. |
| **Missing Step** | The reasoning omits a crucial intermediate step needed to arrive at the correct conclusion. |
| **Coherency** | The logical flow is disrupted due to contradictions or a lack of narrative consistency. |
| **Commonsense** | Lacks real-world knowledge known to the world. |
| **Arithmetic** | Mistakes in performing numerical operations. |

## 3.2 SAE Setup

We use a pre-trained Sparse Autoencoder (SAE) along with SAELens—an interpretability tool designed for analyzing feature directions learned by the SAE. In previous work, SAEs have been shown to reduce superposition by mapping internal activations to more interpretable dimensions. Here, we take the hidden activations from Gemma-2b's CoT generation process (just before the final output layer) as the input to the SAE.

1. **Activation Collection**

   (a) For each of the 1,000 CoT samples, we capture hidden activations at a specific layer (e.g., near the output layer).

   (b) We normalize or batch-process these activations as needed for the SAE.

2. **Sparse Autoencoder**

   (a) The SAE is pre-trained on a broader set of LLM activations.

   (b) We do not fine-tune the SAE on our 1,000-sample set; instead, we use it to generate a sparse feature representation for each CoT sample's activation vector.

3. **Feature Extraction with SAELens**

   (a) SAELens outputs a sparse vector per sample, indicating the magnitude of each sparse feature direction.

   (b) These vectors represent an interpretable decomposition of Gemma-2b's internal reasoning activations.

## 3.3 Correlation Analysis

After extracting the SAE feature vectors, we perform the following steps:

1. Label each sample with the error categories from error taxonomy.

2. Apply correlation measures between error labels and the presence (or magnitude) of each sparse feature direction.

3. Investigate whether certain feature directions consistently co-activate with specific error types. For example, a particular feature direction may align with arithmetic mistakes, indicating a structured relationship between the learned representations and error types.

4. Perform cluster analysis on latent representations using unsupervised learning techniques such as k-means, assessing whether different reasoning failures naturally separate within the sparse encoding space.

5. Conduct an attribution analysis using feature importance techniques to identify which sparse dimensions are the strongest predictors for each error type. This helps determine whether certain feature directions generalize across multiple instances of errors.

6. Train classifiers on the SAE feature vectors to predict the presence of specific error categories, comparing classification performance against raw embeddings to assess whether sparsification enhances interpretability.

# 4 Experimentation and Preliminary Work

Since our study is in its early stages, we focus on data preparation and feature extraction to establish a foundation for further analysis.

## 4.1 Model Configuration

We use **Gemma 2B**, a Large Language Model (LLM) with 2 billion parameters, to generate step-by-step answers for each of the 1,000 GMS8K questions.

## 4.2 IMPLEMENTATION DETAILS

1. Manually classify errors with a small team of trained annotators.
2. Capture activations at a consistent layer across the 1,000 CoT responses.
3. Feed these activations into our pre-trained SAE/SAELens.
4. Collect sparse feature representations for each sample.

This preliminary setup enables an initial exploration of the relationship between sparse feature directions and reasoning errors.

## 5 CONCLUSION

In this paper, we propose a preliminary approach for analyzing reasoning errors in chain-of-thought (CoT) prompting using Sparse Autoencoders (SAEs). Our initial analysis suggests that SAEs could help identify structured patterns in reasoning errors by learning sparse representations of intermediate activations. By advancing these methods, we aim to contribute to a deeper understanding of CoT errors and enhance the interpretability of large language models, fostering greater transparency and trust in their reasoning processes.

## 6 ADDRESSING REVIEWERS COMMENTS

Our key contribution lies in bridging sparse autoencoders (SAEs) and error analysis—a direction underexplored in mechanistic interpretability. While prior work focuses on evaluating the correctness of reasoning steps, our approach targets the internal activation patterns underlying these errors. We intend to introduce a novel methodology to establish feasibility, demonstrating how SAEs can uncover structured relationships between sparse feature directions and reasoning failures. Though empirical validation remains future work, our framework provides a critical foundation for interpretable debugging of chain-of-thought errors.

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
