# OpenReview forum: "Finding Sparse Autoencoder Representations Of Errors In CoT Prompting"
_ICLR.cc/2025/Workshop/BuildingTrust — BuildingTrust_

### Official Review · Reviewer_V2hK · 2025-03-02
**Applying Sparse Autoencoders to Uncover Error Patterns in Chain-of-Thought Reasoning**

**Rating:** 6
**Confidence:** 2

**Review:**

The paper "Finding Sparse Autoencoder Representations of Errors in CoT Prompting" presents a novel methodology for analysing reasoning errors in Large Language Models (LLMs) during chain-of-thought (CoT) generation using Sparse Autoencoders (SAEs). The authors manually labelled 1,000 CoT responses from the GSM8K dataset into nine distinct error categories and extracted SAE features from Gemma-2B's activations. Their aim was to uncover structured patterns in reasoning failures.The work bridges the gap between mechanistic interpretability and practical debugging of LLMs. It focuses on error-specific feature directions in activation space rather than solely on final model outputs.

Strengths

The study's primary strength lies in its innovative application of SAEs to the understudied problem of reasoning error analysis.While SAEs have been explored in mechanistic interpretability, their use for categorizing CoT errors represents a fresh direction with practical implications for improving model transparency. The integration of ROSCOE's error taxonomy provides a structured framework for labelling, and the three-step methodology (error labeling, SAE activation extraction, and correlation analysis) is logically coherent and reproducible.The interdisciplinary approach—combining error taxonomies from prior work with SAE tools like SAELens—demonstrates effective synthesis of existing techniques. Furthermore, the emphasis on intermediate reasoning steps, as opposed to final conclusions, is in alignment with the mounting demand for interpretability in AI systems.

Weaknesses

Notwithstanding the encouraging framework, the paper is currently lacking in empirical validation, thus leaving key questions unanswered. These include the question of whether SAE features reliably correlate with error types, and how they compare to alternative methods. The absence of quantitative results means that the paper's conclusions are not supported by evidence.The sample size of 1,000 CoT responses may be insufficient to capture rare error types (e.g. arithmetic mistakes), and the manual labelling process introduces potential annotation bias, which is not mitigated by inter-annotator agreement metrics. The paper does not provide sufficient technical details about the SAE architecture (e.g. sparsity constraints, training data, layer selection rationale), which limits the ability of the reader to reproduce the results. Furthermore, the paper does not benchmark SAE-based analysis against simpler baselines like linear probes or PCA, leaving its comparative advantage unclear.

Originality and Significance

The work is original in its objective of mapping error categories to sparse feature directions, a direction that has received comparatively little attention in the context of interpretability research. While ROSCOE and related studies focus on evaluating correctness, this study explicitly links internal activations to reasoning failures, thus offering a pathway for targeted debugging. If validated, this approach has the potential to significantly enhance tools for diagnosing and mitigating CoT errors, thereby contributing to more reliable and transparent LLMs.

Clarity

The paper is generally well-structured, with a clear motivation and methodology. However, there is a lack of depth in the technical sections (e.g. SAE setup, correlation analysis) to support replication.Critical components, such as Table 1 (error definitions), are referenced but missing from the submitted text, which disrupts readability.Additionally, the discussion of SAE training and activation normalization is overly brief, leaving ambiguities about implementation choices.

---

### Official Review · Reviewer_TBEM · 2025-03-02

**Rating:** 5
**Confidence:** 3

**Review:**

# Summary
The paper proposes a novel approach for diagnosing internal chain-of-thought reasoning errors in large language models. By leveraging sparse autoencoder representations, the authors extract interpretable features from model activations. They correlate these sparse features with error types from manual annotations, establishing a framework for systematic CoT error analysis and interpretability improvements.
# Strengths
The methods and experiment procedures are clearly described.
# Weakness and Questions
1. Would you mind sharing any initial experimental results?
2. Could you kindly elaborate on how you evaluate the effectiveness of your method, including the specific metrics or the testing procedure?

---

### Official Review · Reviewer_MiSp · 2025-03-03
**This study remains exploratory rather than providing actionable insights for improving LLM trust.**

**Rating:** 4
**Confidence:** 3

**Review:**

The paper touches on an important topic—trust and reliability in LLM reasoning steps—but falls short in execution. While the use of SAEs for interpretability is an interesting direction, the study lacks empirical rigor, making it hard to assess the method’s effectiveness. The authors fail to show that SAEs provide meaningful decompositions of reasoning errors. They also do not compare against other interpretability techniques. To make this work more compelling, future research should also test whether identified sparse directions can actually be used for model correction.

---

### Decision · Program_Chairs · 2025-03-02

Accept